# Effects of Diethyl Phosphate, a Non-Specific Metabolite of Organophosphorus Pesticides, on Serum Lipid, Hormones, Inflammation, and Gut Microbiota

**DOI:** 10.3390/molecules24102003

**Published:** 2019-05-24

**Authors:** Fangwei Yang, Jinwang Li, Guofang Pang, Fazheng Ren, Bing Fang

**Affiliations:** 1Beijing Advanced Innovation Center for Food Nutrition and Human Health, College of Food Science and Nutritional Engineering, China Agricultural University, Beijing 100083, China; fwyang@cau.edu.cn (F.Y.); sdlijinwang@sina.com (J.L.); 519-02@cau.edu.cn (G.P.); renfazheng@cau.edu.cn (F.R.); 2Chinese Academy of Inspection and Quarantine, Beijing 100176, China; 3Key Laboratory of Functional Dairy, Co-Constructed by Ministry of Education and Beijing Government, and Beijing Laboratory of Food Quality and Safety, China Agricultural University, Beijing 100083, China

**Keywords:** endocrine system, hormones, inflammation, microbiome, DNA sequencing

## Abstract

Organophosphorus pesticides (OPs) can be metabolized to diethyl phosphate (DEP) in the gut environment, which may affect the immune and endocrine systems and the microbiota. Correlations between OPs and diseases have been established by epidemiological studies, mainly based on the contents of their metabolites, including DEP, in the serum or urine. However, the effects of DEP require further study. Therefore, in this study, adult male rats were exposed to 0.08 or 0.13 mg/kg DEP for 20 weeks. Serum levels of hormones, lipids, and inflammatory cytokines as well as gut microbiota were measured. DEP significantly enriched opportunistic pathogens, including *Paraprevotella*, *Parabacteroides*, *Alloprevotella*, and *Helicobacter*, leading to a decrease in interleukin-6 (IL-6). Exposure to the high dose of DEP enriched the butyrate-producing genera, *Alloprevotella* and *Intestinimonas*, leading to an increase in estradiol and a resulting decrease in total triglycerides (TGs) and low-density lipoprotein cholesterol (LDL-C); meanwhile, DEP-induced increases in peptide tyrosine‒tyrosine (PYY) and ghrelin were attributed to the enrichment of short-chain fatty acid-producing *Clostridium sensu stricto 1* and *Lactobacillus*. These findings indicate that measuring the effects of DEP is not a proxy for measuring the effects of its parent compounds.

## 1. Introduction

Organophosphorus pesticides (OPs) are frequently detected in food [1,2,3], and about 75% of all registered OPs are metabolized in the body into measurable dialkyl phosphate metabolites, such as diethyl phosphate (DEP) [4]. Human biomonitoring of OP exposure could include the determination of these non-specific metabolites in urine [5,6,7]. Previously, epidemiological studies established correlations between OPs and endocrine system diseases [8,9,10] or infertility [11,12] based on urinary measurements of DEP. The endocrine-disrupting effects of OPs, including chlorpyrifos [13,14,15,16,17,18], diazinon [19,20], malathion [21], triazophos [22], and dimethoate [23], have only been proven in animal studies, which reported a disorder of hormones involved in the hypothalamic‒pituitary‒adrenal [20], hypothalamic‒pituitary‒thyroid [14,17,18,19], and hypothalamic‒pituitary‒gonadal axes [14,15,16,17,21,22,23]. All of these OPs metabolize to DEP in vivo; however, we do not know whether these effects are induced by the parent OPs or the metabolites, especially the non-specific metabolite, DEP.

Ingested OPs are absorbed and metabolized in the gastrointestinal tract and subsequently interact with other organs and tissues due to the existence of enzymes that can metabolize OPs into DEP, including cytochrome P450 isoforms, carboxylesterase, and esterase paraoxonase-1 [24,25]. Therefore, the gut is the first system in the body to be exposed to and affected by pesticides and metabolites. As increasing evidence has emerged about the relationship between gut health and diseases [26,27], recent risk assessments of pesticides have begun to focus on the effects on gut endocrine cells [28], the gut barrier [29,30,31], gut inflammation [32,33], and gut microbiota [32,34,35,36,37]. Combined with the existence of DEP in foods [38], the gut is likely to be influenced by both the OPs and DEP. However, risk assessments of DEP are lacking, especially regarding the effects on gut microbiota and the endocrine system. Thus, in this study, we evaluated the effects of DEP on hormones and gut microbiota in an attempt to clarify whether it is appropriate to use DEP as a marker for OPs when assessing health risks.

## 2. Results

### 2.1. Effects of DEP Exposure on Body Weight and Food Composition

The mean body weight and food consumption of rats in each group during the experimental period are shown in Figure 1. DEP was given in equal molar doses to the 1/500 of the lethal dose 50 (LD_50_) of two frequently detected pesticides, triazophos and chlorpyrifos, used in our previous studies, which led to endocrine-disrupting effects. Exposure to a low dose of DEP (DEP-L) did not affect the body weight of rats significantly (*p* > 0.05); however, a high dose (DEP-H group) was associated with lower body weights compared with the control group, a factor that became significant after 20 weeks of exposure (*p* < 0.05, Figure 1A). No significant differences were found in regard to food consumption between groups (*p* > 0.05, Figure 1B). Furthermore, DEP-induced inhibition of serum acetylcholinesterase (AChE) activity occurred (Figure 2), and DEP did not exhibit the same neurotoxicity as the OPs at the same molar dose (Appendix A).

### 2.2. Effects of DEP Exposure on Serum Levels of Hormones and Lipids

Serum levels of sex and adrenal hormones and lipids in rats exposed to DEP are shown in Figure 3. DEP exposure only increased the concentration of estradiol, leaving other hormones unchanged (*p* > 0.05, Figure 3A,B). Meanwhile, DEP also significantly decreased the serum levels of total triglycerides (TGs) and low-density lipoprotein cholesterol (LDL-C) (*p* < 0.05, Figure 3C).

### 2.3. Effects of DEP on Gut Endocrine Cells and the Immune System

The serum levels of gut hormones and inflammatory cytokines in rats exposed to DEP are shown in Figure 4. The serum concentrations of peptide tyrosine‒tyrosine (PYY) and ghrelin increased significantly in the DEP-H group (*p* < 0.05, Figure 4A). DEP exposure at both doses decreased the serum level of interleukin-6 (IL-6) significantly (*p* < 0.05, Figure 4B).

### 2.4. Effects of DEP on Gut Microbiota

Figure 5 displays the principal coordinate analysis (PCoA) plots. The presence of significant differences in their relative abundance of genera between groups was analyzed using the Student’s *t*-test. As shown in Figure 5A, there was less similarity in genera between DEP groups compared with between treatments and the control. Furthermore, the DEP-H group had less in common with the control group than the DEP-L group, which was in accordance with the idea that the genera composition was less altered in the DEP-L group (Figure 5C).

DEP-L exposure significantly enriched the relative abundance of the *[Bacteroides] pectinophilus* group, while *Paraprevotella*, and *Adlercreutzia* and depleted the concentrations of *Ruminococcaceae_UCG-004*, *Jeotgalicoccus*, and *Faecalibaculum* (Figure 5B, *p* < 0.05). DEP-H increased the relative abundance of *Lactobacillus*, *Parabacteroides*, *Alloprevotella*, *Clostridium sensu stricto 1*, *Helicobacter*, *[Eubacterium] ventriosum* group, *Intestinimonas*, and *norank f Erysipelotrichaceae*, whereas it depleted the concentrations of *Jeotgalicoccus, Ruminococcaceae_UCG-014*, *[Eubacterium] xylanophilum group*, *Candidatus Saccharimonas*, *Defluviitaleaceae UCG-011*, *Catabacter*, *Parasutterella*, *norank f Christensenellaceae*, *unclassified f Peptostreptococcaceae*, *Mucispirillum*, *Erysipelatoclostridium*, and *Candidatus Soleaferrea* (*p* < 0.05, Figure 5C).

## 3. Discussion

The levels of DEP in this study were based on the molar doses corresponding to the doses of triazophos and chlorpyrifos in our previous studies. Triazophos and chlorpyrifos are pesticides frequently detected in vegetables and fruits, and we evaluated the endocrine-disrupting effects of the two pesticides at the dose of 1/500 LD50 (0.164 mg/kg b.w. and 0.3 mg/kg b.w. for triazophos and chlorpyrifos, respectively) [39,40]. Furthermore, the low dose of DEP (0.08 mg/kg b.w.) in this study was also corresponded to the data in the urine of Chinese [41] after being transformed into rats with an average weight of 66.2 kg [42] and a urine volume of 2000 milliliters per day [43].

Previous epidemiologic studies established an association between OPs and disturbed hormone concentrations, especially reproductive hormones, based on urinary DEP measurements [11,12,44]. However, in accordance with Meeker et al. [45], this study could only provide evidence for an inverse relationship between DEP exposure and the serum level of estradiol (Figure 3A). Insufficient levels of estrogen in post-menopausal women have been shown to lead to an increase in plasma TG, and transdermal estradiol treatment can significantly lower plasma TG levels [46,47]. This effect has also been proven in adipocytes [48] and was reported to be the reason for decreased TC and LDL-C in estrogen-treated men [49], which is in accordance with the decreased serum levels of TG and LDL-C found in our study (*p* < 0.05, Figure 3C). Evidence has also been found for the contribution of OPs to the development of obesity and diabetes, which may be due to the specific metabolites of OPs instead of the non-specific DEP.

Previous studies on the endocrine-disrupting mechanism of OPs focused on acetylcholine [50,51] and/or the interaction with hormone receptors [13,52]. In our study, no inhibitory effect of DEP on the activity of AChE was found (Figure 2), and the enriched *Adlercreutzia* in the DEP-L group was shown to be negatively correlated with anxiety behavior [53]. *Adlercreutzia* was also found to be depleted in patients with enteritis [54,55] and multiple sclerosis [56]; these results indicate the detoxification of OPs. However, when we continued to increase the dose of DEP, there were negative effects, such as significantly higher levels of estradiol (Figure 3A, *p* < 0.05), PYY, and ghrelin (Figure 4A, *p* < 0.05).

Gut health, determined by the function of gut endocrine and gut microbiota, influences the physiology and pathophysiology of the host [57]. Gut hormones, including ghrelin, PYY, GIP, GLP-1, and PP, are secreted by the intestinal epithelial cells [58] and communicate between the gut and the brain to affect the metabolism [59,60] and behavior of the host [61,62] by regulating the central nervous system and gut microbiota.

The alteration of gut microbiota, as well as their metabolites, such as short-chain fatty acids (SCFAs), has been reported to play a role in the regulation of the testis‒pituitary axis [63] by disrupting steroid hormone synthesis [64]. It has been verified in cells that butyrate acid may increase the level of estradiol [65]. Accordingly, several butyrate-producing genera were significantly enriched in the DEP-H group, including *Alloprevotella* [66] and *Intestinimonas* [67] (Figure 5C, *p* < 0.05). Furthermore, DEP-induced dysbiosis of gut microbiota also resulted in changes in gut hormones and inflammatory cytokines (Figure 4) [58,68,69]. SCFA-producing gut bacteria were also found to affect the secretion of PYY and ghrelin [70,71], and the enrichment in SCFA-producing *Alloprevotella* [66], *Clostridium sensu stricto* 1 [72], and *Lactobacillus* [73] may have been the reason for the significant increases in PYY and ghrelin in the DEP-H group (Figure 4A, *p* < 0.05). 

DEP exposure at both doses decreased the serum level of IL-6 significantly (Figure 4B, *p* < 0.05), which is a pleiotropic cytokine implicated in acute phase responses and that controls antibody production from B lymphocytes [74]. The decreased IL-6 may be related to the significantly enriched opportunistic pathogens, including *Paraprevotella* [75,76], in the DEP-L group (Figure 5B, *p* < 0.05) and *Parabacteroides* [75,77], *Alloprevotella* [77,78], and *Helicobacter* [79,80] in the DEP-H group (Figure 5C, *p* < 0.05). IL-6 is implicated in autoimmune and immunoinflammatory diseases such as SLE [81], rheumatoid arthritis [82], diabetes [83] and cancer [84]. It is necessary in future studies to aim at the mechanism of DEP in reducing IL-6 while leaving TNF-α unchanged, which may help to design new inhibitors of IL-6. Furthermore, future studies may be of interest to evaluate the impact of DEP on other cytokines of the innate immune system such as IL-1, IL-12, IL-18 and IL-23. 

In conclusion, chronic exposure to DEP, a non-specific metabolite of OPs, affected the gut microbiota, serum hormones, and proinflammatory cytokines in rats, with stronger responses observed at high doses. We speculate that the endocrine-disrupting effects of OPs depend more on their specific metabolites, but too much DEP would also disturb serum estradiol levels. These findings suggest that DEP is not a viable marker for precursor OPs when evaluating their toxicity on the endocrine system.

## 4. Materials and Methods

### 4.1. Animals and Experimental Design

Seven-week-old male Wistar rats were purchased from the experimental animal center of Weitong Lihua Laboratory Animal Technology Company (Beijing, China). DEP (CAS 598-02-7) with a purity above 97% was purchased from Huaxia Regent Company (Chengdu, China). Chemicals were dissolved in dimethyl sulfoxide (DMSO) and then diluted with 0.9% saline. The final DMSO concentration was 0.1%. All procedures were performed humanely at China Agricultural University, following protocols approved by the university’s ethics committee.

After a one-week acclimation period, rats were randomly divided into three groups (*n* = 10 per group) to ensure minimal differences in bodyweight between groups. Rats from each group were housed in multiple cages to minimize the cage effect. DEP was administered by gavage once daily for 20 weeks at a dose of 0.08 (DEP-L) or 0.13 mg/kg (DEP-H) bodyweight. The administered doses were calculated on the basis of weekly weights. Rats in the control group were administrated with a vehicle of 0.9% saline containing 0.1% DMSO (control). The animals were given the same standard chow diet, and both feed and distilled water were available ad libitum. Throughout the experimental period, all rats were housed in cages and kept in a single room at 24 ± 2 °C, 40% to 70% relative humidity, and a 12 h light/dark cycle. Bodyweight and food consumption were measured weekly throughout the experiment.

At the end of the exposure period, rats were anesthetized for blood collection from the orbit and decapitated. Plasma was centrifugated at 5000× *g* for 20 min at 4 °C after standing for 2 h at 37 °C. The hemolyzed plasma was then discarded, and the upper layer of the serum was transferred to new sterile tubers and stored at −80 °C until further analysis.

### 4.2. Fecal Sample Collection and Bacterial DNA Extraction

Fecal samples were collected the day before sacrifice and later stored in sterile tubes containing RNA at −20 °C. Bacterial genomic DNA was extracted using the phenol‒chloroform extraction method [85]. DNA was quantified using a NanoDrop spectrophotometer (OneC, Thermo Fisher Scientific, Waltham, MA, USA) and stored at −80 °C until further analysis.

### 4.3. AChE Activity Measurements

Serum activity of AChE in rats at the end of exposure was determined spectrophotometrically using an AChE assay kit (Life Technologies, Carlsbad, CA, USA, Cat. No. #A12217), according to the manufacturer’s instructions. The inhibition rate was calculated from the activity value of the control subjects and represented as a percentage.

### 4.4. Serum Lipid Profiles

Serum levels of total cholesterol (TC), TG, LDL-C, and high-density lipoprotein cholesterol (HDL-C) were measured at the core laboratory of Peking University Third Hospital using an automatic biochemical analyzer.

### 4.5. Hormones and Cytokine Measurements

Serum levels of insulin, glucagon, pancreatic polypeptide (PP) (total), glucagon-like peptide 1 (GLP-1) (active), gastric inhibitory polypeptide (GIP) (total), PYY (total), ghrelin (active), leptin, IL-6, monocyte chemoattractant protein-1 (MCP-1), and tumor necrosis factor-α (TNF-α) were quantified using a multiplexing kit (Milliplex MAP Rat Metabolic Hormone Magnetic Bead 13-Plex Panel, Catalog Number: MMHMAG-44K, RMHMAG-84K, Merck Millipore, Burlington, MA, USA) according to the manufacturer’s protocol. Briefly, antibody beads were mixed together in a bottle and reagent solutions, which include standard, quality control, and blank solutions, were prepared according to the kit’s guidelines. Samples and the standard quality controls were added into the well, then 200 µl of the assay buffer was added. After shaking for 10 min at room temperature, 25 µl of serum matrix solution and 25 µl of mixed antibody beads were successively added to the background. The plate was sealed and incubated at 4 °C overnight, followed by washing three times with wash buffer. In total, 50 µl of the detection antibodies were added and incubated for 30 min at room temperature on a plate shaker. Then, 50 µl of streptavidin-phycoerythrin was added and incubated for 30 min at room temperature followed by the removal of all well contents and washing 3 times with the washing buffer. After that, 100 µl of Luminex Sheath Fluid (which was used to drive the samples to the optic component of the Bio-plex machine) was added to each well before putting the plate inside the Bio-Plex machine to read the absorption.

Serum levels of luteinizing hormone (LH), follicle-stimulating hormone (FSH), adrenocorticotropic hormone (ACTH), testosterone, estradiol, adrenaline, and corticosterone were determined using the radioimmunoassay (RIA) method of Beijing Sino‒UK Institute of Biological Technology (Beijing, China). Briefly, for measurement of serum LH, 50 μL samples were assayed using a LH kit, following the manufacturer’s instructions, and the assay sensitivity was 0.41 mIU/mL. For measurement of serum FSH, 50 μL samples were assayed using an FSH kit, which applies solid-phase RIA using an FSH-specific antibody immobilized to the wall of a polypropylene tube, and the assay sensitivity was 0.57 mIU/mL. For measurement of serum ACTH, 100 μL samples were assayed using an ACTH kit, which uses antibody-coated tubes. For measurement of serum testosterone and estradiol, 25 μL samples were assayed using a total testosterone kit and a total estradiol kit, respectively, following the manufacturer’s instructions, and the assay sensitivity was 0.15 ng/mL and 1.61 pg/mL, respectively. For measurement of serum adrenaline, 25 μL samples were assayed using an adrenaline kit, which applies solid-phase RIA using an adrenaline-specific antibody immobilized to the wall of a polypropylene tube. For measurement of serum total corticosterone, 50 μL aliquots were assayed using a total corticosterone kit, following the manufacturer’s instructions. In all analyses, the intra-assay coefficients of variation were below 10% and there were three parallel wells for each sample.

#### 4.6. S rRNA Gene Sequencing

DNA was first amplified using the universal primers, B341 (5ʹ-CCTACGGGNGGCWGCAG-3ʹ) and B785R (5ʹ-GACTACHVGGGTATCTAATCC-3ʹ), to target the V3 and V4 regions of bacterial 16S rRNA. PCR amplification was performed according to the protocol of the kit (KAPA Biosystems, Wilmington, MA, USA) with conditions as follows: Initial 95 °C for 3 min, followed by 25 cycles of 95 °C for 30 s, 55 °C for 30 s, and 72 °C for 30 s, and then a final extension of 72 °C for 5 min. After purifying the positive amplicons using AMPure XP beads (AGENCOURT AMPURE XP Kit, Beckman Coulter Inc., Brea, CA, USA), the amplicons were assessed and quantified using a Qubit fluorometer (Thermo Fisher Scientific, San Jose, CA, USA) and KAPA Library Quantification Kit (KAPA Biosystems) according to the manufacturer’s instructions. The resulting products were pooled and sequenced on an Illumina Miseq PE300 platform (Illumina, San Diego, CA, USA) using a paired-end sequencing strategy (2 × 300 bp). Raw data were spliced and filtered by removing the merged sequencing reads ≤200 bp or ≥500 bp in length and reads with a quality score of ≤20 or those containing ≥4 ambiguous base calls using MOTHUR software (version 1.35.1). The filtered and trimmed high-quality reads were then used to select the operational taxonomic units (OTUs) with USEARCH software (version 7.0) and a default cutoff of 97% sequence similarity. The OTUs were further subjected to the Ribosomal Database Project classifier software for taxonomic identification. An 80% confidence threshold was used for taxonomic assignment, and the taxonomic assignment was achieved at different levels, including phylum, class, order, family, and genus.

### 4.7. Gut Microbiota Analysis

β-diversity was assessed using weighted UniFrac distance matrices, which was the basis of PCoA and the abundance heatmap [86]. Differences in the composition of gut microbiota were further assessed by nonparametric tests using Metastats software [87]. Differences in the gut microbiome composition between the control and the DEP-treated group were further assessed by the Student’s *t*-test using the FDR as the correction for multiple testing [88].

### 4.8. Statistical Analysis

For body weight and hematological indices (serum lipids, serum levels of hormones, and cytokines), data were expressed as mean ± SD and analyzed using one-way ANOVA, followed by the Duncan’s multiple range test at a 95% confidence interval (SPSS version 19.0, IBM SPSS, Armonk, NY, USA).

## Figures and Tables

**Figure 1 molecules-24-02003-f001:**
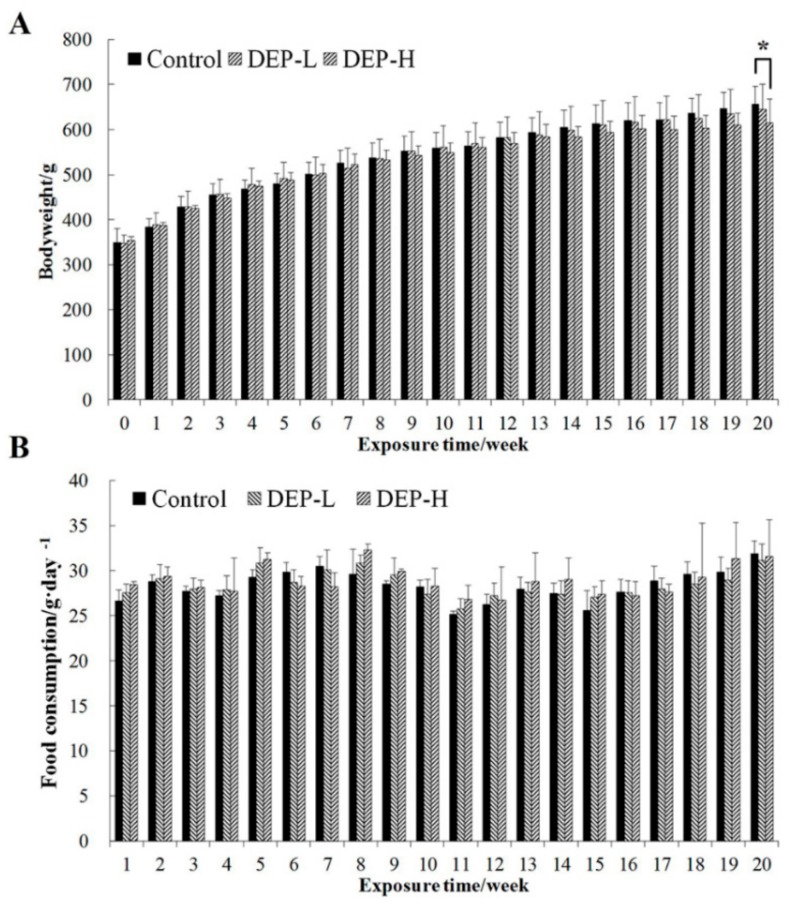
Body weight (**A**) and food consumption (**B**) of rats during the experimental period. Data are expressed as mean ± standard deviation. Control, vehicle treatment; a low dose of diethyl phosphate (DEP-L), 0.08 mg/kg body weight; a high dose of diethyl phosphate (DEP-H), 0.13 mg/kg body weight. The symbol * indicates significant differences from the corresponding control group at the confidence interval of 95%.

**Figure 2 molecules-24-02003-f002:**
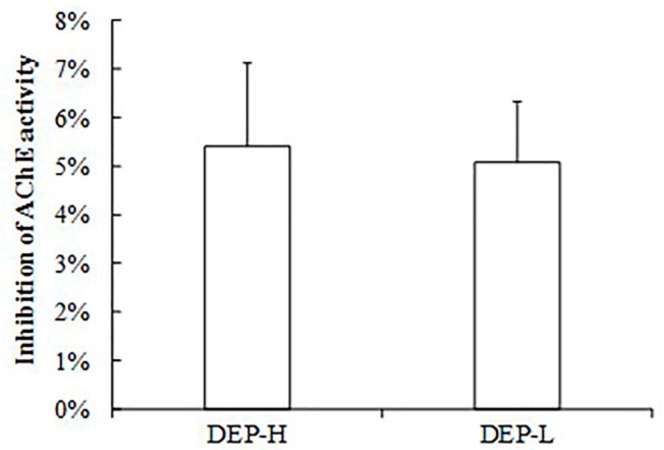
Inhibition in the activity of acetylcholin esterase (AChE) in the serum of rats exposed to DEP compared with the control group. The inhibition rate was calculated as the percentage of the activity value in DEP groups divided by the value in the control group.

**Figure 3 molecules-24-02003-f003:**
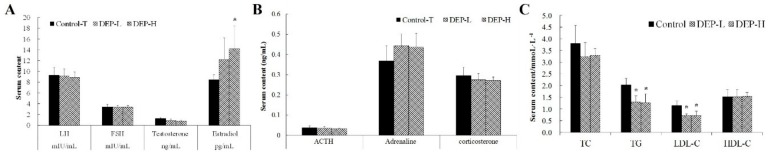
Serum levels of sex hormones (**A**) and adrenal hormones (**B**) and blood lipid profiles (**C**) in rats. The symbol * indicates significant differences from the corresponding control group at the confidence interval of 95%. LH, luteinizing hormone; FSH, follicle-stimulating hormone; ACTH, adrenocorticotropic hormone; TC, total cholesterol; HDL-C, high density lipoprotein cholesterol.

**Figure 4 molecules-24-02003-f004:**
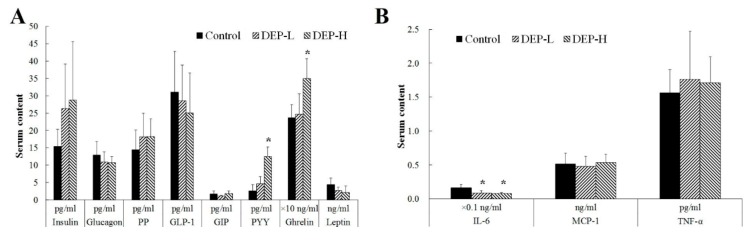
Serum levels of gut hormones (**A**) and inflammatory cytokines (**B**) in rats. The symbol * indicates significant differences from the corresponding control group at the confidence interval of 95%. PP, pancreatic polypeptide; GLP-1, glucagon-like peptide 1; GIP, gastric inhibitory polypeptide; PYY, peptide tyrosine tyrosine; IL-6, interleukin 6; MCP-1, monocyte chemoattractant protein 1; TNF-α, tumor necrosis factor alpha.

**Figure 5 molecules-24-02003-f005:**
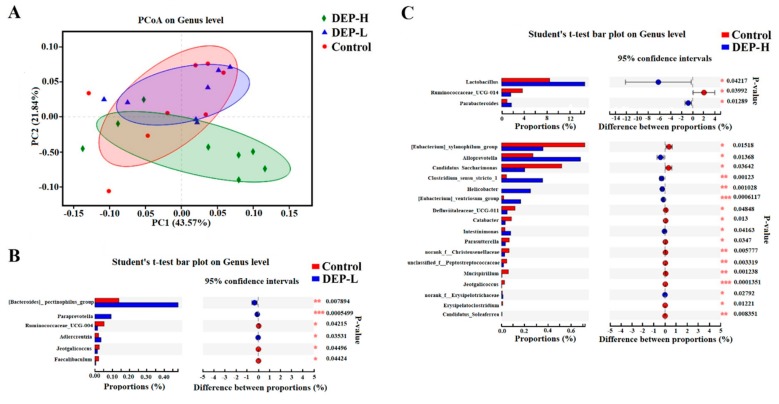
Altered gut microbiome community structure induced by DEP. (**A**) Principal coordinate analysis (PCoA) plots of gut microbiome patterns (circles, control; triangles, DEP-L; diamonds, DEP-H). The percentage of variation explained by the plotted principal coordinates is indicated on the axes. (**B**, **C**) Significantly altered genera analyzed by the Student’s *t*-test in the DEP-L and DEP-H groups, respectively, compared with the control.

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
