# Peer review of "Effects of Diethyl Phosphate, a Non-Specific Metabolite of Organophosphorus Pesticides, on Serum Lipid, Hormones, Inflammation, and Gut Microbiota"

_molecules, 2019, doi:10.3390/molecules24102003_

Round 1

Reviewer 1 Report

I have no further comments.  

Author Response

Thank you very much for your enlightening suggestions and comments!

Reviewer 2 Report

The effects of diethyl phosphate (DEP), the metabolite of organophosphorus pesticides (OPs), were investigated in rats exposed for 20 weeks. I like to give the following comments.

1.      The dose of DEP is important and is essential to introduce in clear. DEP at 0.08 mg/kg as low but 0.13 mg/kg as high which needs the reference(s).

2.      In Figure 2, it seems not difference between low dose and high dose in reduction of AChE activity. Why?

3.      In Figure 4, assay of GLP-1 is unclear. Is the total GLP-1 or the active GLP-1 determined? Additionally, the used kits for hormone and cytokines were unclear.

4.      Gut microbiota analysis needs reference(s) to support. It is a novel view of this report.

5.      SCFA-producing gut bacteria became the main target of DEP. Why? Please give a speculation.

6.      Food intake was not changed by DEP. Is it reasonable?

Author Response

1. The dose of DEP is important and is essential to introduce in clear. DEP at 0.08 mg/kg as low but 0.13 mg/kg as high which needs the reference(s).

Thank you for your advice and we added these information in the revised manuscript (lines 118-124). The levels of DEP were based on the molar doses corresponding to the doses of triazophos and chlorpyrifos in our previous studies. Triazophos and chlorpyrifos are pesticides frequently detected in vegetables and fruits, and we evaluated the endocrine-disrupting effects of the two pesticides at the dose of 1/500 LD50 (0.164 mg/kg b.w. and 0.3 mg/kg b.w. for triazophos and chlorpyrifos, respectively).

Furthermore, the low dose of DEP (0.08 mg/kg b.w.) in this study was also corresponded to the data in urine of Chinese.

A clinical study conducted by Fu and Xie (2013) found that the highest concentration of DEP in urine was 442.69 μg/L in adults of Dongguan City, Guangdong Province, China. And the exposure dose in rats was calculated as below according to the Food and Drug Administration guidelines (Food and Drug Administration, 2005):

Rat Dose (mg/kg bw) = Human Equivalent Dosage (mg/kg bw) × [Human K/ Animal Km] = 442.69 μg/L × 2.0 L/66.2 kg bw × (37/6) = 13.37 μg/kg bw. × 6.2 = 82.89 μg/kg bw ≈ 0.08 mg/kg bw.

In which, Human Km = 37 and Rat Km = 6; according to the Report on Nutrition and Chronic Diseases of Chinese Residents (2015) issued by the National Health Commission of the People's Republic of China, the average weight of Chinese adult males is 66.2 kg (National Health Commission of the People's Republic of China, 2015). The normal range for 24-hour urine volume is 800 to 2,000 milliliters per day (National Library of Medicine - National Institutes of Health, 2019).

References

The references which pointed out the LD50 dose of triazophos (JMPR, 2002; Jain et al., 2011) and chlorpyrifos (Wang et al., 2009; Mansour and Mossa Abdel-Tawab, 2010) in rat was listed as below:

Joint FAO/WHO Meeting on Pesticide Residues (JMPR), Pesticide residues in food - 2002 - Joint FAO/WHO Meeting on Pesticide Residues, TRIAZOPHOS 2002. http://www.inchem.org/documents/jmpr/jmpmono/2002pr14.htm

Jain, S., Ahmed, R.S., Arora, V.K., Banerjee, B.D., 2011. Biochemical and histopathological studies to assess chronic toxicity of triazophos in blood, liver and brain tissue of rats. Pesticide Biochemistry and Physiology 100(2), 182–186.

Wang, H.P., Liang, Y.J., Long, D.X., Chen, J.X., Hou, W.Y., Wu, Y.J., 2009. Metabolic profiles of serum from rats after subchronic exposure to chlorpyrifos and carbaryl. Chemical Research in Toxicology 22, 1026–1033.

Mansour, S.A., Mossa Abdel-Tawab, H., 2010. Oxidative damage, biochemical and histopathological alterations in rats exposed to chlorpyrifos and the antioxidant role of zinc. Pesticide Biochemistry and Physiology 96(1), 14–23.

The references for the data when calculating the dose was listed as below:

Fu, Y., Xie, W., 2013. Analysis of exposure levels of several common environmental pollutants in medical examination population. Journal of Qiqihar University of Medicine, 34(19): 2897-2898.

Food and Drug Administration, 2005. Guidance for industry: estimating the maximum safe starting dose in initial clinical trials for therapeutics in adult healthy volunteers. Center for Drug Evaluation and Research (CDER), 7.

National Health Commission of the People's Republic of China, 2015. The Report on Nutrition and Chronic Diseases of Chinese Residents (2015), 30 June 2015. http://www.nhc.gov.cn/jkj/s5879/201506/4505528e65f3460fb88685081ff158a2.shtml

National Library of Medicine - National Institutes of Health, 2019. Urine 24-hour volume, 22 March 2019. https://medlineplus.gov/ency/article/003425.htm

2. In Figure 2, it seems not difference between low dose and high dose in reduction of AChE activity. Why?

First of all, DEP did not influence the activity of AChE significantly at both of the doses, and no clinical signs of cholinergic toxicity, such as tremor, salivation or diarrhea, were observed in any animal during the study.

OPs were reported to inhibit AChE activity, which was mainly due to the group of the “-P=O” (see the below figure in detail). The leaving group portion of the OP (indicated by R3 in the figure) associates with the anionic site on glutamate 334 followed by the binding of the phosphate portion of the OP to the hydroxyl group on serine 203 in AChE. Because DEP does not have a leaving group, DEP cannot bind first with glutamate 334, and thus does not further bind to serine 203 in AChE.

Reference

Chambers, J.E., Carr, R. L., (2002). Acute toxicities of organophosphates and carbamates. In: Massaro, E.J. (Ed.), Handbook of Neurotoxicology, Vol. 1. Totowa, NJ, Humana Press Inc., pp. 3-16.

3. In Figure 4, assay of GLP-1 is unclear. Is the total GLP-1 or the active GLP-1 determined? Additionally, the used kits for hormone and cytokines were unclear.

Sorry for the unclear description. We measured the active GLP-1 and we specified in the revised version (line 206). At the same time, we described the procedure of the method in detail in the method part (lines 210-221, 225-238).

4. Gut microbiota analysis needs reference(s) to support. It is a novel view of this report.

Thank you for your advice and we added several references in the revised manuscript (lines 260-261, 263, 494-500).

5. SCFA-producing gut bacteria became the main target of DEP. Why? Please give a speculation.

Yes, our results suggested that DEP exposure affect the relative abundance of SCFA-producing gut bacteria in rats. Gut microbiota regulate the physiology and pathophysiology of the host though the metabolites including SCFAs [Alonso et al. 2014; Lyte 2013], previous studies also indicate that gut microbiota play important roles in the regulation of lipid and cholesterol metabolism [Lecomte et al., 2015] by affecting the content of SCFA. Maybe it was not DEP targeted SCFA-producing bacteria, instead, the disturbance in gut microbiota resulted in changes in SCFAs. which was reported to play a role in the regulation of the testis‒pituitary axis [64] by disrupting steroid hormone synthesis [65]. We speculated this in the discussion part (lines 142-151).

References

Alonso, C., Vicario, M., Pigrau, M., Lobo, B., Santos, J., Intestinal barrier function and the brain−gut axis. In: Lyte M, Cryan JF (eds) Microbial endocrinology, the microbiota−gut−brain axis in health and disease. Springer, New York, 2014, pp 73–96.

Lyte, M., Microbial endocrinology in the microbiome–gut–brain axis: how bacterial production and utilization of neurochemicals influence behavior. PLOS Pathogens 2013, 9(11), e1003726.

Lecomte, V., Kaakoush, N.O., Maloney, C.A., Raipuria, M., Huinao, K.D., Mitchell, H.M., Morris, M.J., 2015. Changes in gut microbiota in rats fed a high fat diet correlate with obesity–associated metabolic parameters. PLoS One 10(5), e0126931.

Al-Asmakh, M.; Stukenborg, J. B.; Reda, A.; Anuar, F.; Strand, M. L.; Hedin, L.; Pettersson, S.; Soeder, O., The gut microbiota and developmental programming of the testis in mice. Plos One 2014, 9(8), e103809.

Gérard, P., Gastrointestinal tract, microbial metabolism of steroids. Timmis KN (ed) Handbook of hydrocarbon and lipid microbiology 2010.

6. Food intake was not changed by DEP. Is it reasonable?

Yes, we believed that it was reasonable that there was no significant difference in food intake between the experimental group and the control group rats. DEP did not exhibit the cholinergic toxicity as the parent OPs and no clinical signs of cholinergic toxicity were observed in any animal during the study. Furthermore, there were also studies reported even the exposure of OPs did not affect the food intake of rats, such as diazinon [Adamkovicova et al., 2016], chlorpyrifos [Fang et al., 2018; Reygner et al., 2016], triazophos [Sharma and Sangha, 2016], all of which can be metabolized and degraded to non-specific metabolite DEP in vivo.

References

Adamkovicova, M., Toman, R., Martiniakova, M., Omelka, R., Babosova, R., Krajcovicova, V., ... & Massanyi, P. (2016). Sperm motility and morphology changes in rats exposed to cadmium and diazinon. Reproductive Biology and Endocrinology, 14(1), 42.

Fang, B., Li, J. W., Zhang, M., Ren, F. Z., & Pang, G. F. (2018). Chronic chlorpyrifos exposure elicits diet-specific effects on metabolism and the gut microbiome in rats. Food and Chemical Toxicology, 111, 144-152.

Reygner, J., Lichtenberger, L., Elmhiri, G., Dou, S., Bahi-Jaber, N., Rhazi, L., ... & Abdennebi-Najar, L. (2016). Inulin supplementation lowered the metabolic defects of prolonged exposure to chlorpyrifos from gestation to young adult stage in offspring rats. PLoS One, 11(10), e0164614.

Sharma, D., & Sangha, G. K. (2016). Effects of glucosinolates rich broccoli extract against triazophos induced toxicity in Wistar rats. A randomized control study. Journal of Biomedical Science, 5(4), 25.

This manuscript is a resubmission of an earlier submission. The following is a list of the peer review reports and author responses from that submission.

Round 1

Reviewer 1 Report

Major points:

1. If OPs presumably would be metabolized into DEP by different organs/tissues, then gut system should be primarily exposed to OPs rather than DEP. Thus with the in vivo study, why not give OPs to reveal the direct in vivo impact of pesticide?

2. While give DEP, why chose 0.08 and 0.13 mg/kg as low and high? What are the rationales? Comparable to the exposure in humans?

3. There are quite several parameters measured and the rationales for doing such experiments were not given to link to the pathology or pathophysiology of OPs/DEP in the body.

4. In many figures, the units of the determined molecules in the x-axes did not indicate any meaningful terms.

5. Only male rats were included. Especially it was mention in the discussion section of a study in post-menopausal women….; if females animals could be used in parallel, it may be more intriguing to clarify the impacts on sex/steroid hormones in this study.

Minor points:

1. There are language problems to be improved.

2. What is PYY? Why it is important to be examined?

Reviewer 2 Report

The study was designed to investigate the impact of metabolites of OPs specifically DEP on levels of sexual hormones, pituitary hormones, inflammatory cytokines as well as composition of gut microbiota.   There are several major issues however need to be addressed:

1.      What is the rationale that only male Wistar rats were used in the study? Is there evidence indicating gender differences in response to OPs exposure?

2.       Lack of an additional control group. The authors should include a group of rats treated with the parent compound (OPs). It is critical when comparing the inhibitory effect of DEP on AChE using OP as a control as well. This group is a must to include.

3.      No Figure 3C was provided in the manuscript.

4.      While ANOVA was used to compare body weight and food intake, it is unclear what statistical method(s) were used to compare other endpoint (i.e in figure 3 A and B, was a student t-test used to compare DEP-H with control?)  

5.      No concentrations of DEP were measured in serum as well as feces to correlated the levels with various endpoints reported in the study.

6.       Student t-test for PCOA analysis is inappropriate for multiple groups.

7.      For relative abundance, why DEP-L treatment enriched the relative abundance of [Bacteroides]_pectinophilus_group, while DEP-H did not?

8.      What is the rationale to choose the levels of DEP used in the experiments?

9.      Daily oral gavaging for 20 weeks may induce stress, so affect the dynamic profile of hormones and inflammatory cytokines.  Gavaging is not an ideal dosing route for this study.

10.  English editing is strongly recommended, for example:

 Ln 135: “there was “bad” effects such as…”

 On numerous places in the manuscript, the authors used “content” instead of “concentrations” or “levels” to refer to hormone measurement results.  

Ln 137: “There were also evidences about the contribution of OPs in the development of obesity and diabetes “a”, which…”

Ln 121:” It was found that lack of estrogen in post-menopausal women…” did it mean postmenopausal women have no estrogen?